# Evaluating the Landslide Stability and Vegetation Recovery: Case Studies in the Tsengwen Reservoir Watershed in Taiwan

**Chun-Hung Wu**

Department of Water Resources Engineering and Conservation, Feng Chia University, Taichung 40724, Taiwan; chhuwu@fcu.edu.tw; Tel.: +886-424-517-250-3223

**Abstract:** The sediment yield from numerous landslides triggered in Taiwan's mountainous regions by 2009 Typhoon Morakot have had substantial long-term impacts on the evolution of rivers. This study evaluated the long-term evolution of landslides induced by 2001 Typhoon Nari and 2009 Typhoon Morakot in the Tsengwen Reservoir Watershed by using multiannual landslide inventories and rainfall records for the 2001–2017 period. The landslide activity, vegetation recovery time, and the landslide spatiotemporal hotspot analyses were used in the study. Severe landslides most commonly occurred on 35–45° slopes at elevations of 1400–2000 m located within 500 m of the rivers. The average vegetation recovery time was 2.29 years, and landslides with vegetation recovery times exceeding 10 years were most frequently retrogressive landslide, riverbank landslides in sinuous reaches, and the core area of large landslides. The annual landslide area decline ratios after 2009 Typhoon Morakot in Southern Taiwan was 4.75% to 7.45%, and the time of landslide recovery in the Tsengwen reservoir watershed was predicted to be 28.48 years. Oscillating hotspots and coldspots occupied 95.8% of spatiotemporal patterns in the watershed area. The results indicate that landslides moved from hillslopes to rivers in the 2001–2017 period because the enormous amount of sediment deposited in rivers resulted in the change of river geomorphology and the riverbank landslides.

**Keywords:** landslide evolution; landslide activity; vegetation recovery time; spatiotemporal hotspot

## 1. Introduction

Several serious landslide disaster events were caused by heavy rainfall events between 2001 and 2010, including 2001 Typhoon Toraji [1], 2004 Typhoon Mindulle [2], 2004 Typhoon Aere, and 2009 Typhoon Morakot [3], in the mountainous areas of central and southern Taiwan. The landslide events are believed to be related to the 1999 Chichi earthquake [4]. The return period of heavy rainfall brought by Typhoon Morakot (5–10 August 2009), which caused approximately 45,000 landslides concentrated in central and southern Taiwan, was estimated to be over 200 years [3]. Moreover, loose deposits from the numerous landslides in mountainous areas continues to affect watershed evolution and landslide recovery. As of 2021, mountainous areas in southern Taiwan remain under high risk of landslides and debris flows.

The analysis of landslide recovery or landslide evolution had been widely used in the long-term observation of landslide distribution after large earthquake events, including the 1999 Chichi earthquake [5], 2005 Kashimir earthquake [6], 2008 Wenchuan earthquake [7,8], and 2015 Gorkha earthquake [9]. Landslide evolution in the years following a large earthquake or extreme rainfall event in a watershed with dense landslide cases is the key determinant of watershed management and the mitigation of secondary geohazards [10]. Some artificial factors, including land use [11,12] and road development [13], were the significant factors for the evolution and reoccurrence of landslide. Some studies have been conducted on the characteristics of landslide evolution, including landslide activity [6,7], the spatiotemporal distributions of landslides [8,14], landslide recovery characteristics [15], and landslide evolution trends [10,16], with regard to severe landslides induced by large earthquakes or extreme rainfall. However, few studies have explored landslide evolution

by examining spatiotemporal hotspots. Lin et al. (2017) [17] explained the distribution of landslide hotspots and coldspots on a catchment scale by using multiannual records of landslides, rainfall, and earthquakes for the 2003–2012 period in Taiwan. Conventional hotspot analysis can only describe the clustering pattern and intensity of a particular spatial location for a specific time interval; it cannot explain underlying trends over time [18]. The spatiotemporal evolution of landslide distribution patterns is challenging to investigate but merits scholarly attention.

New analysis methods by using the multi landslide inventories, including spatiotemporal hotspot and landslide activity, had been used to explore the characteristic of landslide evolution after large earthquake or extreme rainfall events [6,7,19]. The emerging hot spot analysis in ArcGIS Pro software has been employed in tracing the spread of COVID-19 [20], locating the source area of pollutant emissions [16], and assessing spatiotemporal changes in fisheries [21]. Emerging hotspot analysis was used in an assessment of long-term landslide evolution in Taiwan after 2009 Typhoon Morakot [19]. The results explained the spatiotemporal pattern and distribution of landslide hotspots and coldspots. Landslide-concentrated areas in the upper reaches of watersheds were determined to have poor landslide recovery and be highly susceptible to further landslides [19]. Emerging hotspot analysis can both explain changes in the spatiotemporal pattern and location of landslide hot spots and evaluate the intensity and location of landslide clustering. Landslide activity data have been employed in assessing the rates of and differences in landslide recovery after large earthquake-induced landslides [6,7] but not after extreme rainfall-induced landslide events.

The present study analyzed characteristics of landslide evolution by assessing landslide activity, estimating vegetation recovery time, and detecting the pattern and spatiotemporal distribution of landslides by using multiannual landslide inventories and long-term rainfall records from 2001 to 2017. Moreover, the characteristics of landslide evolution after large earthquakes were compared with the corresponding characteristics after extreme rainfall events. The Tsengwen Reservoir Watershed (*TRW*) suffered the most serious landslide disasters of all reservoir watersheds in Taiwan following 2009 Typhoon Morakot. The landslide evolution characteristics in the watershed with dense rainfall-induced landslides are poorly understood, but these characteristics represent information essential for watershed management and disaster prevention.

## 2. Research Area

The *TRW* (Figure 1) is a watershed located in the upper reaches of the Tsengwen River in southwestern Taiwan. The average elevation in the *TRW* is 959 m, and the average slope is 29.2°; 49.5% of the total area has a slope of greater than 30°. According to 1/5000 basin geological maps of Taiwan [22], the main stratigraphical formations in the *TRW* (Figure 2 and Table 1) include the Miocene-era Changchihkeng formation, Pliocene-era Ailiaochiao formation, and Miocene- to Pliocene-era Tangenshan sandstone (occupying 34.2%, 16.6%, and 16.4% of the watershed, respectively) [17]. The *TRW* is also surrounded by nine faults (Figure 2). Regarding land use types, forest, agriculture, development, rivers, and bare land account for 80.6%, 10.4%, 1.4%, 4.3%, and 3.3% of the total area based on the land use maps produced in 2008 by National Land Surveying and Mapping Center in Taiwan.

The average (standard deviation) annual precipitation from 2001 to 2017, based on the records of Matoushan Rainfall Station (Figure 1), was 2998.8 (934) mm. Rainfall characteristics in the *TRW* are non-uniform in space and time. The 3 years with the most precipitation were 2005 (4892 mm), 2008 (4118 mm), and 2001 (3764 mm). Those with the least precipitation were 2003 (1533 mm), 2013 (1735 mm), and 2002 (1878 mm; Figure 3). The difference between the precipitation in 2003 and that in 2005 is 1.12 times the average annual precipitation. The total precipitation in the rainy season (May–October; 2749 mm) was 91.7% of the average annual precipitation (2998.2 mm). The precipitation in August 2009, the largest monthly total in the considered period, was 2425 mm—70.2% of the annual precipitation.

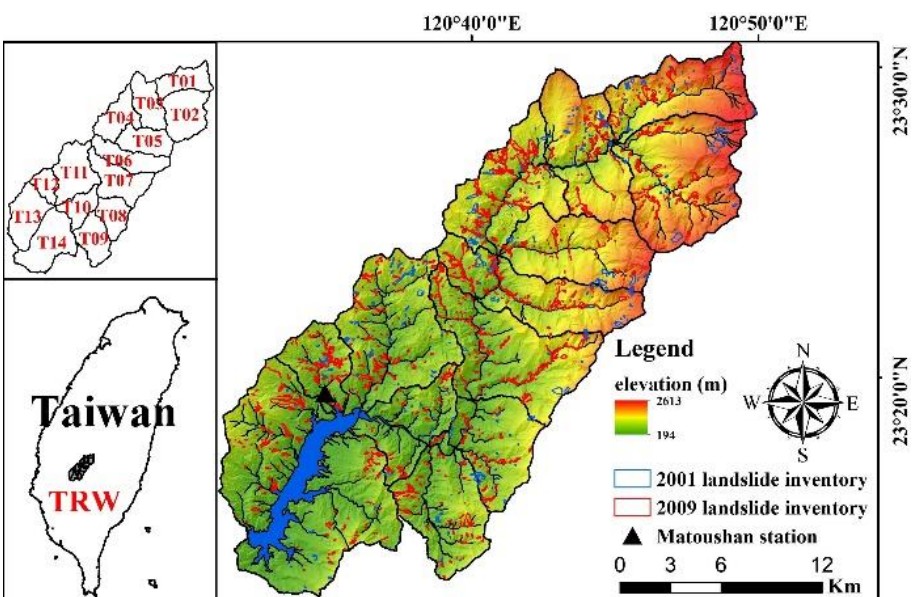

**Figure 1.** The distribution of elevation, rivers, and landslide after 2001 Typhoon Nari and 2009 Typhoon Morakot in the Tsengwen Reservoir watershed (Abbreviated as *TRW*). The down-left figure shows the location of *TRW* in Taiwan, and the up-left figure shows the names of sub-watersheds in the *TRW*.

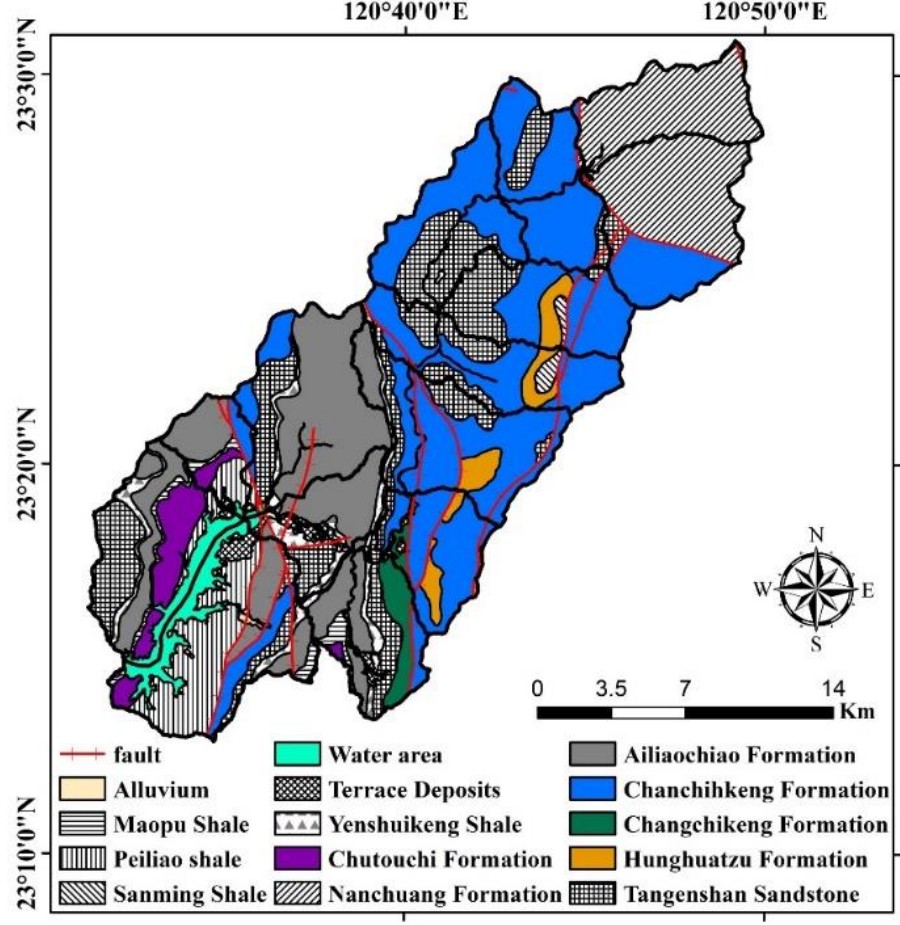

**Figure 2.** The distribution of geological settings and faults in the *TRW*.

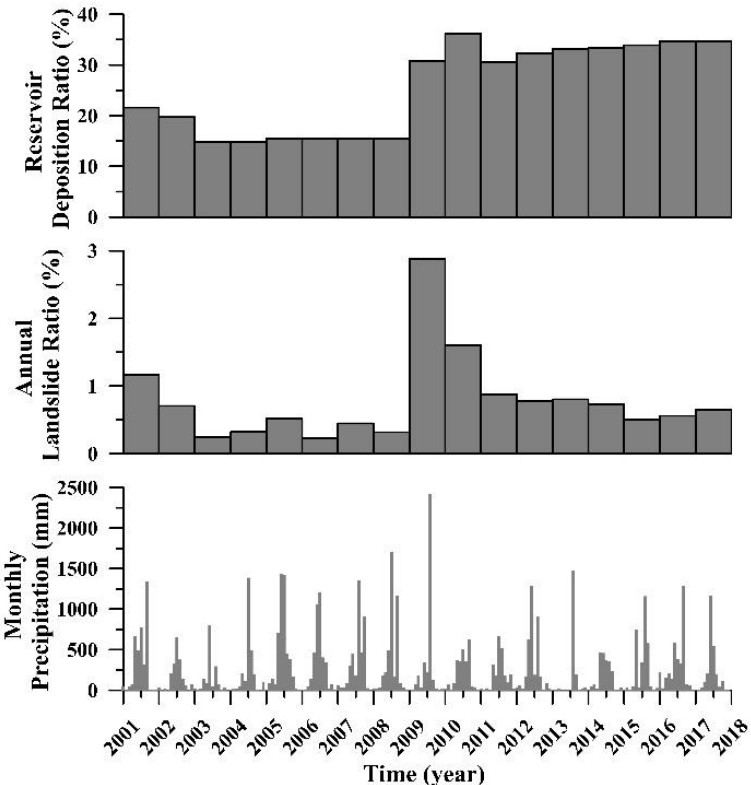

**Figure 3.** The distribution of monthly precipitation, landslide ratio, and reservoir deposition ratio from 2001 to 2017 in the *TRW*. The up, middle, and down figures are the distribution of reservoir deposition ratio, annual landslide ratio, and monthly precipitation from 2001 to 2017 in the *TRW*.

Landslide and debris flow were the two main geohazards in the *TRW*, and the 1999 Chichi earthquake and heavy rainfall events during the 2000s were the main landslide triggers. Typhoon Nari (9–10 September 2001) and Typhoon Morakot (5–10 August 2009) caused the two main rainfall-induced landslide events during the 2000s. In the *TRW*, the accumulated precipitation brought by 2001 Typhoon Nari and 2009 Typhoon Morakot was 990.5 and 2158.5 mm (62.5% of the annual precipitation in 2009), respectively. The landslide ratios (i.e., the ratio of landslide area to watershed area) after 2001 Typhoon Nari and 2009 Typhoon Morakot were estimated to be 1.17% and 2.88%, respectively. Events with a landslide ratio exceeding 1.0% are considered serious disaster events in Taiwan [3]; thus, these two typhoons are regarded as serious disaster events within the *TRW*. The sediment deposition volume in the Tsengwen Reservoir increased by approximately $1.28 \times 10^8$ m$^3$ from 2001 to 2017 (Figure 3). According to the field survey results, the majority of landslides can be classified as shallow landslide [23].

**Table 1.** Statistical data of landslide induced by 2001 Typhoon Nari and 2009 Typhoon Morakot in each stratigraphical division in the *TRW*.

| Stratigraphical Division | Lithology | Occ. Per. (%) | Landslide Ratio (%) | |
| --- | --- | --- | --- | --- |
| | | | In 2001 | In 2009 |
| Water Area | - | 3.07 | 0.00 | 0.27 |
| Alluvium | Gravel, sand, silt, and clay | 0.70 | 0.80 | 1.96 |
| Ailiaochiao Formation | Thin alternation of siltstone and shale | 16.65 | 0.80 | 2.52 |
| Changchihkeng Formation | Sandstone, sandstone interbedded with shale | 34.16 | 1.37 | 3.48 |
| Chutouchi Formation | Muddy sandstone, interbedded muddy sandstone and shale | 2.76 | 0.36 | 5.43 |
| Hunghuatzu Formation | Thick-bedded fine sandstone or siltstone, thick alternation of fine sandstone and siltstone | 2.48 | 0.74 | 4.13 |

**Table 1.** *Cont.*

| Stratigraphical Division | Lithology | Occ. Per. (%) | Landslide Ratio (%) | |
| --- | --- | --- | --- | --- |
| | | | In 2001 | In 2009 |
| Maopu Shale | Shale with thin-bedded sandstone | 0.99 | 0.17 | 5.69 |
| Nanchuang Formation | Alternation of sandstone and shale | 12.44 | 1.93 | 2.91 |
| Peiliao Shale | Shale and sandy shale | 5.58 | 0.34 | 1.00 |
| Sanming Shale | Shale, shale intercalated with thin-bedded siltstone | 0.49 | 4.72 | 5.30 |
| Terrace Deposits | Sand, silt, mud and gravel | 1.19 | 0.02 | 0.00 |
| Tangenshan Sandstone | Thick-bedded massive sandstone and muddy sandstone | 16.39 | 1.45 | 2.65 |
| Yenshuikeng Shale | Massive shale, occasionally intercalated with thin-bedded siltstone | 3.11 | 0.44 | 2.13 |

Note: The Occ. Per. means the occupied percentage of each stratigraphical division area to the watershed area.

## 3. Materials and Method

### 3.1. Materials

A digital elevation model (DEM) and multiannual landslide inventories were the main study materials. The DEM had a spatial resolution of 5 m, and the 5 m × 5 m grid was the basic analysis unit in the study. The landslide inventories, corresponding to the 2001–2017 period, were produced by Taiwan's Forestry Bureau, and landslides were identified from images captured by the Formosat-2 satellite with spatial resolutions of 2–5 m. Multiannual inventories of landslides in the *TRW* (for the years 2001–2017) were examined, and the minimum landslide area in the inventories was 100 m². Landslides with an area of greater than 10 hectares (ha) were considered large landslides. The main types of landslide [24] included in the annual landslide inventories from 2001 to 2017 in Taiwan were fall, slide, and flow [19]. The deep-seated landslide cases, similar to the deep-seated gravitational slope deformation in central Taiwan [25], were not listed in the annual landslide inventories. Data on daily rainfall from 2001 to 2017 were collected from Matoushan Rainfall Station (Figure 1). The *TRW* contains 14 subwatersheds (Figure 1). The subwatersheds in the upper, middle, and lower reaches of the river are labeled as T01–T03, T04–T08, and T09–T14, respectively.

A recovery characteristics comparison of landslides induced by large earthquake and extreme rainfall events was conducted to explore the characteristic of landslide evolution. The large earthquake-induced landslides considered were the landslide cases after the 2005 Kashmir Earthquake (*Mw* = 7.6) in Pakistan [9] and the 2008 Wenchuan Earthquake (*Mw* = 7.9) in China [7,8,16]. The extreme rainfall-induced landslides used in the study were landslides occurring in the Chishan River Watershed [10], Ailiao River Watershed [19], and Taimali River Watershed [19] after 2009 Typhoon Morakot, and the *TRW* before and after 2009 Typhoon Morakot.

### 3.2. Landslide Activity

Landslide activity, which is useful in analyzing the spatiotemporal evolution of landslides, has been assessed by using the presence or absence of landslides in specific years [7,8] or by examining the ratio of the active landslide area to the total landslide area [26]. Following the landslide activity assessment methods used in the evolution discussion of earthquake-induced landslide cases [6,14,27], the criteria for the five types of landslide activity considered are presented in Table 2. The meaning of landslide activity used in a single hillslope scale was the current moving condition of the hillslope [25], but that used in a watershed scale was the vegetation recovery and the stability of loose materials deposited on the hillslopes [6,14,27]. The meanings of extremely active, very active, and active landslide were that the loose material deposited on the hillslope was instability or the vegetation recovery was too poor to stabilize the hillslope. The inventories of landslides in the *TRW* (2001–2017) were appropriate materials for assessing and comparing the activity of landslides induced by 2001 Typhoon Nari and 2009 Typhoon Morakot. The landslide activity induced by the typhoons was evaluated using multiannual landslide inventories for the 2001–2008 and 2009–2017 periods.

**Table 2.** The criteria of landslide activity in the study.

| Activity Type | Criteria |
|---|---|
| Extremely Active | Landslides present in the annual inventories from 2001 to 2008 or from 2009 to 2017. |
| Very Active | Landslides present in the annual inventories from 2005 to 2008 or from 2013 to 2017. |
| Active | Landslides present in the annual inventories of 2007 to 2008, or only in the annual inventory of 2008 for 2001 Typhoon Nari. Landslides present in the annual inventories of 2016 to 2017, or only in the annual inventory of 2017 for 2009 Typhoon Morakot. |
| Dormant | Landslides present in one or more annual inventories of 2001 to 2007 or 2009 to 2016 but absent in the annual inventory of 2008 or 2017. |
| Inactive | Landslides present in the annual inventory of 2001 or 2009 but absent in the annual inventories from 2002 to 2008 or 2010 to 2017. |

*3.3. Landslide Frequency and Vegetation Recovery Time*

Landslide frequency in the study was defined as the total occurrence of landslide identified in each grid from 2001 to 2017. The area that had been identified as landslide at least once from 2001 to 2017 in the study had been named as the landslide-identified area. The total occurrence of landslide-identified from 2001 to 2017 can be considered as the maximum vegetation recovery time (in years). The landslide recovery time had been estimated by using different factors, including landslide density [28], landslide rate [29], number of landslides [30], and landslide area [6]. Vegetation recovery is a significant factor to assess the landslide stability in long-term landslide evolution analysis [31] because the reinforcement of vegetation root contributes to the hillslope stability, especially in the shallow landslide cases [32,33]. The total occurrence of identified landslide of a grid was three, which meant that the grid was identified as landslides three times over the 17-year period. If the three landslide-identified years were discontinuous (e.g., 2005, 2008, and 2012), the vegetation recovery time was considered as 1 year. However, if the three landslide-identified years were continuous (e.g., 2002–2004), the vegetation recovery time was considered as 3 years. Herein, the maximum recovery time was taken as the vegetation recovery time in the study. Vegetation recovery time can be employed as an index to assess the difficulty of vegetation recovery in a watershed and in individual landslide cases.

*3.4. Emerging Hotspot Analysis*

The analysis of landslide spatiotemporal hotspots was based on the landslide spatiotemporal cube (*STC*) model, which represented landslide clustering patterns in each location at various time intervals. The STC model was composed of numerous 5 m × 5 m landslides-identified grids in the TRW from 2001 to 2017 in chronological order. The time series of landslide clustering intensity, represented by the combination of basic units at the same location, described the temporal evolution of landslides.

The emerging spatiotemporal mining method in ArcGIS Pro software was used in the *STC* model to explore the mechanism of temporal landslide evolution in the *TRW*. The Getis-Ord Gi statistic [34] was used to estimate the clustering intensity and to classify hot spot patterns (Table 3). The hotspot classification was based on the landslide clustering intensity in the neighborhood of specific 5 m × 5 m grids in both time and space, and specific basic units were designated as hotspots or coldspots if they featured high and low values of landslide clustering, respectively. Under the emerging spatiotemporal mining method, the consistency and intensity of landslide clustering in each time step was calculated, as was the significance of the autocorrelation and dependence in the vicinity of specific 5 m × 5 m grids. The method also used a space-time implementation of the Mann-Kendall statistic [35] to measure the intensity of landslide clustering in the neighborhood of specific basic units. The time step was set as 1 year, and the neighborhood distance was set as 5, 25, 50, 100, and 200 m for further comparison in the study.

**Table 3.** The classifications and definition of emerging landslide hotspots and coldspots in the study.

| Pattern | Definition |
|---|---|
| Consecutive (CHS or CCS) * | A landslide grid with a single uninterrupted run of statistically significant hotspot or coldspot grids in the final year during the research time period. The landslide grid has never been a statistically significant hotspot or coldspot before the final hotspot or coldspot run. |
| Diminishing (DHS or DCS) | A landslide grid that has been a statistically significant hotspot or coldspot for 90% of the research time period, including the final year. In addition, the clustering intensity of landslides in each year is decreasing (increasing) overall, and that decrease (increase) is statistically significant. |
| Historical (HHS or HCS) | The most recent year is not hotspot or coldspot, but at least 90% of the research time period has been a statistically significant hotspot or coldspot. |
| Intensifying (IHS or ICS) | A landslide grid that has been a statistically significant hotspot or coldspot for 90% of the research time period. In addition, the clustering intensity of landslide for each year increased (decreased) overall, and that increase (decrease) was statistically significant. |
| New (NHS or NCS) | A landslide grid identified as a statistically significant hotspot or coldspot since the first year of the research time period but was not previously identified as a statistically significant hotspot or coldspot. |
| Oscillating (OHS or OCS) | A statistically significant hotspot or coldspot for the final year that has a history of also being a statistically significant coldspot or hotspot during a prior year. Less than 90% of the research time period has been a statistically significant hotspot or coldspot. |
| Persistent (PHS or PCS) | A landslide grid that has been a statistically significant hotspot or coldspot for 90% of the research time period with no discernible trend indicating an increase or decrease in the clustering intensity of landslide over time. |
| Sporadic (SHS or SCS) | A landslide grid that is an on-again then off-again hotspot or coldspot. Less than 90% of the research time period has been a statistically significant hotspot or coldspot, and none of the time-step intervals have been a statistically significant coldspot or hotspot. |
| No pattern detected (No) | The analysis grid does not fit any definition of hotspot or coldspot classifications |

*: CHS and CCS are the abbreviations of consecutive hotspot and consecutive coldspot. The regulation of abbreviation was applied to every hotspot and coldspot pattern in the study.

## 4. Results

### 4.1. Multiannual Rainfall and Landslide Data

The Matoushan Rainfall Station (Figure 1) was used as the representative rainfall station in the *TRW*. The average annual rainfall recorded by the station from 2001 to 2017 (2998 mm) is 1.06 times that of the corresponding level from 1969 to 2020 (2835 mm) [36]. Table 4 presents data on daily and annual rainfall and on annual landslides in the *TRW* from 2001 to 2017. The annual rainfall distribution in the *TRW* was clearly non-uniform in time. The difference between the lowest annual rainfall (1533 mm in 2003) and the highest annual rainfall (4892 mm in 2005) was 3359 mm, which is 1.12 times the average annual rainfall. The daily rainfall corresponding to return periods of 1.11, 2, 5, 10, 20, 50, 100, and 200 years, based on station records from 1969 to 2020, were estimated to be 144.5, 300.6, 447.9, 545.4, 639.0, 760.1, 850.9, and 941.3 mm, respectively [33]. The maximum daily rainfall from 2001 to 2008 of 545.5 mm was observed twice, once during 2001 Typhoon Nari and once during 2005 Typhoon Haitang, and the return period was estimated to be 10–20 years. The maximum daily rainfall during Typhoon Morakot of 1206.0 mm was the highest amount of rainfall recorded at Matoushan Rainfall Station. The return period of daily rainfall from 2010 to 2017 was <10 years, and the maximum daily rainfall of 502.5 mm was recorded during 2015 Typhoon Soudelor.

The landslide area in 2001 (565.2 ha) was the highest for 2001–2008, and that in 2009 (1392.9 ha) marked the historical high recorded in the *TRW*. The average annual landslide area corresponding to the 2009–2017 period was 2.25 times that of the 2001–2008 period. The annual average number of landslides from 2009 to 2017 was 2.29 times larger than that from 2001 to 2008.

Southern Taiwan suffered serious landslide disaster during 2009 Typhoon Morakot, and the *TRW* was located in this region. The number of landslides in 2009 was 5.3 times larger than that in 2008. The number of the landslide cases with area greater than 100,000 m$^2$, 1000–100,000 m$^2$, and <1000 m$^2$ in 2009 were 29, 1114, and 7, respectively. The average landslide length/width ratio in all landslide cases induced by 2009 Typhoon Morakot was 7.2, and this information showed that the majority type of landslide was rainfall-triggered slide.

**Table 4.** The statistical data of annual rainfall and landslide from 2001 to 2017 in the *TRW*.

| Year | Annual Rainfall (mm) | Daily Rainfall (mm) | Return Period (Years) | Landslide Area (ha) | Landslide Number |
|------|----------------------|---------------------|------------------------|---------------------|-------------------|
| 2001 | 3764 | 545.5 | 10 to 20 | 565.2 | 324 |
| 2002 | 1878 | 96.5 | <1.11 | 375.3 | 241 |
| 2003 | 1533 | 233.0 | 1.11 to 2 | 117.4 | 194 |
| 2004 | 2593 | 502.5 | 5 to 10 | 154.4 | 294 |
| 2005 | 4892 | 545.5 | 10 to 20 | 252.1 | 375 |
| 2006 | 3774 | 436.5 | 5 to 10 | 108.0 | 169 |
| 2007 | 3917 | 267.0 | 1.11 to 2 | 216.0 | 255 |
| 2008 | 4112 | 501.5 | 5 to 10 | 153.5 | 217 |
| 2009 | 3454 | 1206.0 | >200 | 1392.9 | 1150 |
| 2010 | 2492 | 379.5 | 2 to 5 | 775.6 | 1091 |
| 2011 | 2225 | 241.5 | 1.11 to 2 | 422.4 | 434 |
| 2012 | 3569 | 433.0 | 2 to 5 | 375.5 | 513 |
| 2013 | 1736 | 381.0 | 2 to 5 | 388.3 | 565 |
| 2014 | 2077 | 172.5 | 1.11 to 2 | 352.0 | 461 |
| 2015 | 3050 | 502.5 | 5 to 10 | 240.8 | 345 |
| 2016 | 3478 | 491.5 | 5 to 10 | 270.4 | 407 |
| 2017 | 2428 | 334.0 | 2 to 5 | 312.6 | 425 |

Note: The daily rainfall meant the maximum daily rainfall in the specific year, and the return period was based on the daily rainfall.

### 4.2. Landslide Activity

Landslide activity data provide insights into the stability of rainfall-induced landslides for the analysis of the spatial distribution of landslide evolution. Figure 4 presents the spatial distribution of landslide activity after 2001 Typhoon Nari and 2009 Typhoon Morakot in the *TRW*. Areas of extremely active, very active, active, dormant, and inactive landslides after 2001 Typhoon Nari constituted 0.04, 40.8, 277.9, 939.5, and 561.4 ha, respectively. The corresponding areas after 2009 Typhoon Morakot were 54.0, 82.0, 424.7, 1577.0, and 762.2 ha, respectively. The accumulated area of extremely active, very active, and active landslides after 2009 Typhoon Morakot was 1.76 times that after 2001 Typhoon Nari, indicating that the landslides induced by Typhoon Morakot were more difficult to recover. The extremely active landslide area in subwatersheds in the upper reaches of the *TRW* increased substantially from 0 ha after 2001 Typhoon Nari to 42 ha after 2009 Typhoon Morakot, reflecting the difficulty of landslide recovery in the upstream subwatersheds.

A comparison of landslide activity after the 2005 Kashmir Earthquake [6] and 2009 Typhoon Morakot revealed that the proportions of unstable (extremely active, very active, or active) landslides and stable landslides (dormant or inactive landslides) were 28% and 72% in 2018 (13 years after the 2005 Kashmir earthquake). However, the corresponding proportions 8 to 9 years after typhoons in the TRW were 17.5–19.3% and 80.7–82.5%, respectively. These results indicate that vegetation recovery of landslides induced by 2009 Typhoon Morakot in the TRW was easier than vegetation recovery of landslides induced by 2005 Kashimir earthquake in Pakistan [6].

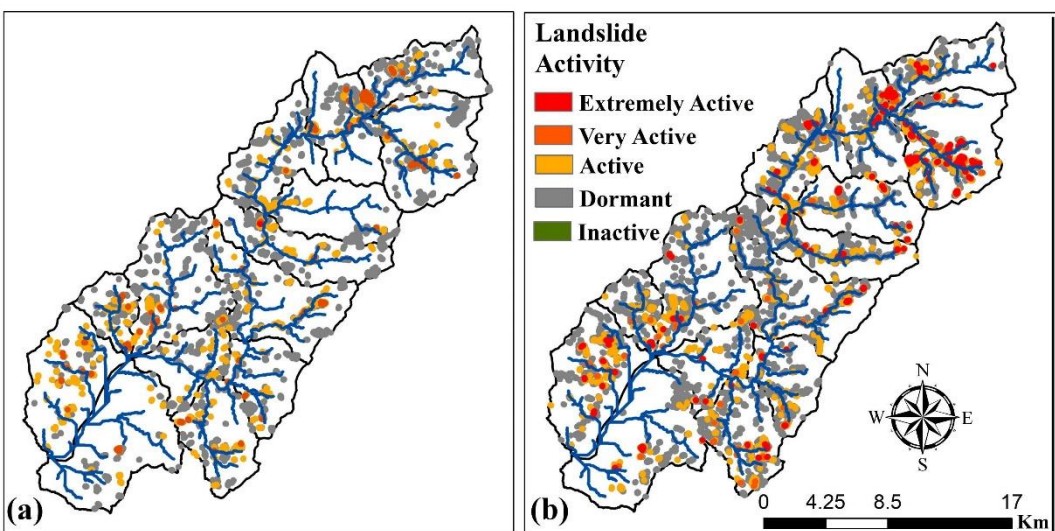

**Figure 4.** The distribution of landslide activity in the *TRW*. (**a**) is the distribution of landslide activity after 2001 Typhoon Nari and (**b**) is after 2009 Typhoon Morakot.

*4.3. Relationship between Landslide Ratio and Geomorphological Factors*

The relationship between earthquake-induced landslide area and geomorphological control factors, including elevation, slope, and distance to the river, had been discussed in analyses of the evolution of earthquake-induced landslides [6–8,18]. The temporal distribution of landslide ratios with respect to elevation, slope, and distance to the river from 2001 to 2017 in the *TRW* is shown in Figure 5. The landslide ratio distribution for the 2001–2007 period was used to observe the evolution of landslides induced by 2001 Typhoon Nari, and that for the 2009–2017 period was used to examine the evolution of landslides induced by 2009 Typhoon Morakot.

The landslide ratios were concentrated at 1800 to 2200 m elevation intervals after 2001 Typhoon Nari. However, from 2003 to 2007, they were concentrated at 800–1400 m elevation intervals. The landslide ratios at elevations below 1000 m and above 1600 m decreased notably during this period. After 2009 Typhoon Morakot, the landslide ratios were concentrated at 600 m to 1400 m elevation intervals. From 2010 to 2017, they were concentrated at 1200 m to 1800 m elevation intervals. The landslide ratios at elevations below 1200 m decreased notably, and those at elevations above 1800 m exhibited small changes from 2010 to 2017.

Only the distribution of extremely active landslides after 2009 Typhoon Morakot is plotted herein (Figure 5) because the area of extremely active landslides after 2001 Typhoon Nari was quite small (0.04 ha). Extremely active landslides were clustered at 1400 to 2000 m elevation intervals after Typhoon Morakot. The extremely active or very active landslides were clustered at moderately high elevations after large earthquakes. For example, they were clustered from 1000 to 1600 m after the 2005 Kashmir Earthquake [6] and below 2000 m after the 2008 Wenchuan Earthquake [7]. Overall, the clustered area and locations of extremely active or very active landslides after large earthquakes or extreme rainfall events was very similar.

The landslide ratios were concentrated at two slope intervals, 35° to 45° and 55° to 60°, after 2001 Typhoon Nari and 2009 Typhoon Morakot. The landslide ratios were clustered at the same intervals from 2003 to 2007 and from 2010 to 2017. The extremely active landslides were concentrated at 35–45° slope intervals after 2009 Typhoon Morakot. Extremely active or very active landslides were also clustered at 30–40° after the 2005 Kashmir Earthquake [6] and at 30° to 50° after the 2008 Wenchuan Earthquake [7].

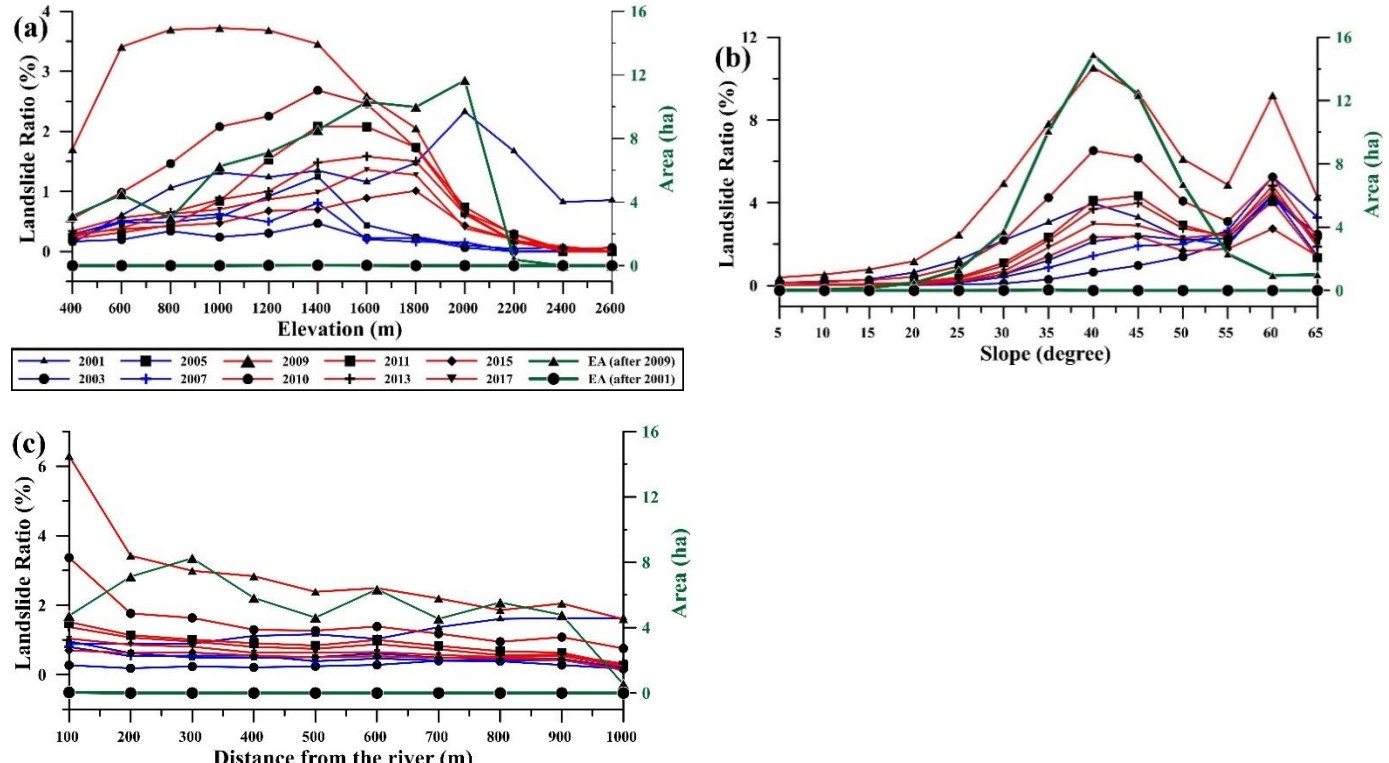

**Figure 5.** The relationship between landslide ratios and geomorphological factors, including elevation in (**a**), slope in (**b**), and distance to the rivers in (**c**), from 2001 to 2017 in the *TRW*. The EA line (green) means the extremely active landslide after 2009 Typhoon Morakot.

In Taiwan, loose debris from landslides after heavy rainfall events was generally deposited in rivers and gullies [10,15]. The *TRW* was divided into 10 buffer areas with 100 m distance intervals from the rivers. The landslide ratios in all river buffer intervals after 2001 Typhoon Nari ranged from 0.88% to 1.63%, but those after 2009 Typhoon Morakot were all greater than 1.86%. The highest landslide ratio in all river buffer intervals after 2001 Typhoon Nari was 1.63% within 1000 m of the river, and that after 2009 Typhoon Morakot was 6.3% within 100 m of the river.

Extremely active landslides were clustered along the fault line after the 2005 Kashmir Earthquake [6] but were concentrated along rivers in the *TRW* after 2009 Typhoon Morakot. This indicates that large pieces of loose material from landslides were transported by excess surface runoff and flood discharge and deposited in the rivers, thereby inducing riverbank landslides and change of river geomorphology.

*4.4. Landslide Frequency and Vegetation Recovery Time*

The distribution of landslide frequency from 2001 to 2017 in the *TRW* is shown in Figure 6. The total landslide-identified area was 26.4 km$^2$ (5.5% of the total area), and the average landslide frequency was 2.29. The area with landslide frequency = 1 constituted 16.28 km$^2$, which was 61.8% of the landslide-identified area. The area with landslide frequency greater than 10 constituted 0.394 km$^2$. Those landslides were most commonly retrogressive landslide or riverbank landslides in the sinuous reaches and the core area of the large landslide cases.

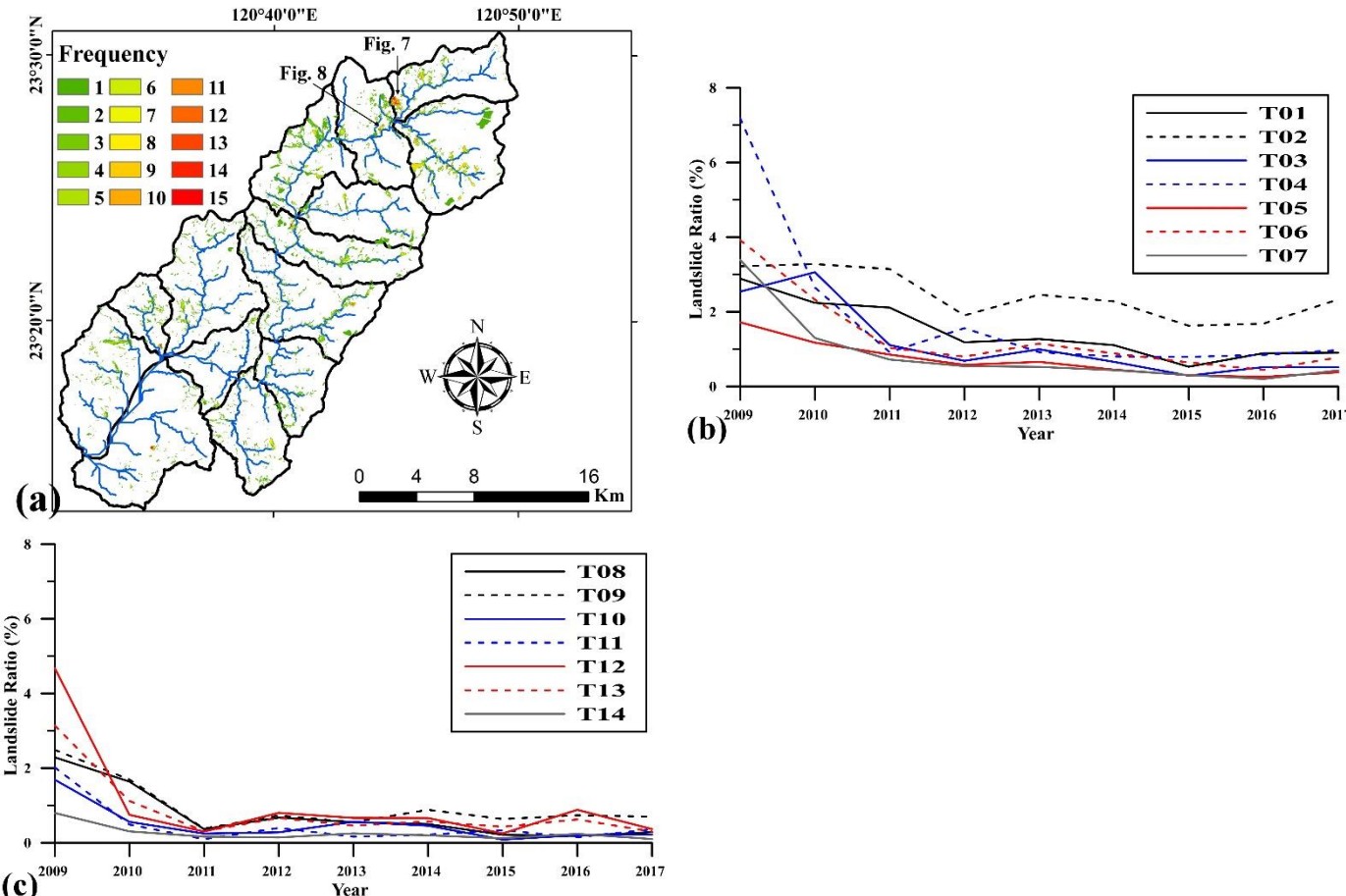

**Figure 6.** The landslide frequency from 2001 to 2017 (**a**) and the landslide ratio in each subwatershed from 2009 to 2017 (**b**,**c**) in the *TRW*.

The landslide frequency in the subwatersheds in the upper reaches of the *TRW* (2.28–3.73) differed substantially from the distributions in the subwatersheds in the middle and lower reaches (1.82–1.97 and 1.72–2.35, respectively). The average vegetation recovery time in the subwatersheds in the upper reaches was 1.41 to 1.59 times longer than that in the subwatersheds in the middle and lower reaches. The subwatersheds in the upper reaches had the highest landslide ratio (Figure 6b,c) and smallest catchment area, indicating that the large amount of sediment yield from numerous landslides was deposited in the narrow upper reaches, and that resulted in the change of river flowing path and the riverbank landslides.

Large landslides and riverbank landslides in the sinuous reaches had the longest vegetation recovery time of all landslides in the *TRW*. The 2009 landslide inventory contained 22 large landslide cases, the mean vegetation recovery time of which was 3.29 years. Two large landslide cases with mean vegetation recovery times of greater than 6 years occurred in the sinuous upper reaches of the subwatersheds. The large landslide occurring in a gully source area in the T01 subwatershed (Figure 7) can be a typical example to explain the vegetation recovery condition. The area and relief of the large landslide were 35.7 ha and 503.2 m, and the average slope was 42.1°. The stratigraphical formation and lithology of the large landslide were the Nanchuang formation and the sandstone and shale. The activity type of the large landslide was very active. The mean vegetation recovery time of the large landslide was 10.25 years. The vegetation recovery time in the boundary area of the large landslide was less than 3 years, indicating that landslides were easily re-induced in this boundary area. However, the vegetation recovery time in the core area of the large

landslide case was over 7 years, and vegetation recovery in the core area of the large landslide was difficult.

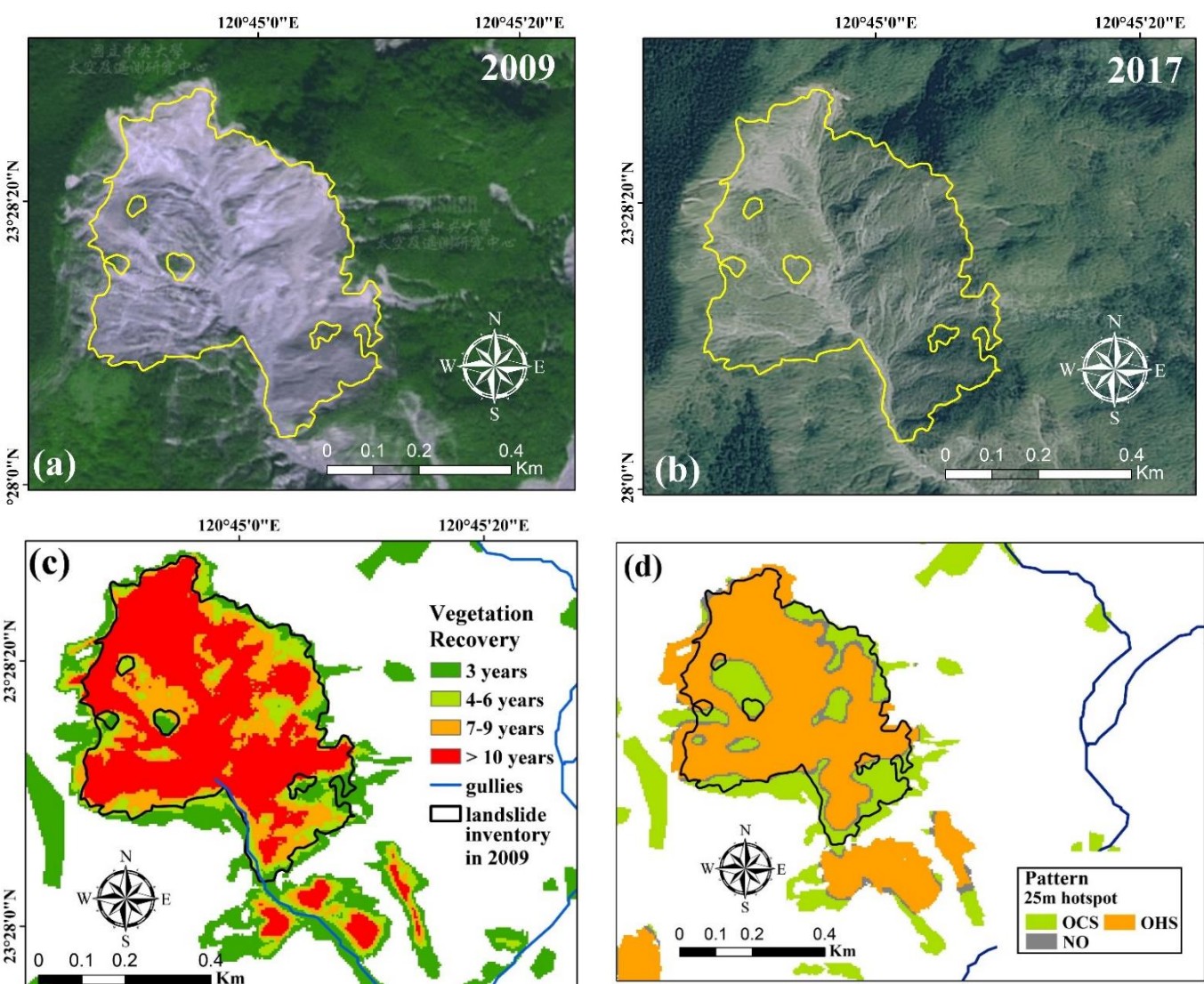

**Figure 7.** The aerial images in 2009 (**a**) and 2017 (**b**), the distribution of vegetation recovery time (**c**), and the spatiotemporal hotspots (**d**) in the large landslide in the T01 subwatersheds (Figure 6).

The river buffer area was defined as the area within 300 m of the river. The landslide-identified area occupied 7.5% of the total area in the river buffer area in the *TRW*, and the mean vegetation recovery time in the river buffer area was 2.65 years. The proportion of the landslide-identified area and the landslide frequency in the river buffer area were slighter larger than those in the *TRW*. The enormous amount of loose material from landslides was deposited in the confluence of river waters and sinuous reaches, resulting in riverbank landslides, especially in the upper reaches of landslide-concentrated subwatersheds. The riverbank landslide occurring in the confluence downstream of the T01 and T02 subwatersheds (Figure 8) is a typical example demonstrating the impacts of large sediment deposits in sinuous reaches on landslide recovery. The area and relief of the large landslide were 0.92 ha and 269.7 m, and the average slope was 32.5°. The stratigraphical formation and lithology of the large landslide were the Changchihkeng formation and the sandstone and shale. The activity type of the large landslide was active and very active. The riverbank landslide was located in a sinuous reach with the sinuosity index = 1.62, and the sediment deposition depth from 2009 to 2011 in this sinuous reach was estimated as 4.32 m.

The mean vegetation recovery time of the riverbank landslide (Figure 8) was 6.79 years (2.56 times longer than that in the *TRW*). The considerable amount of sediment deposition in the river in the vicinity of the riverbank landslide resulted in the formation of river meanders and several new riverbank landslides.

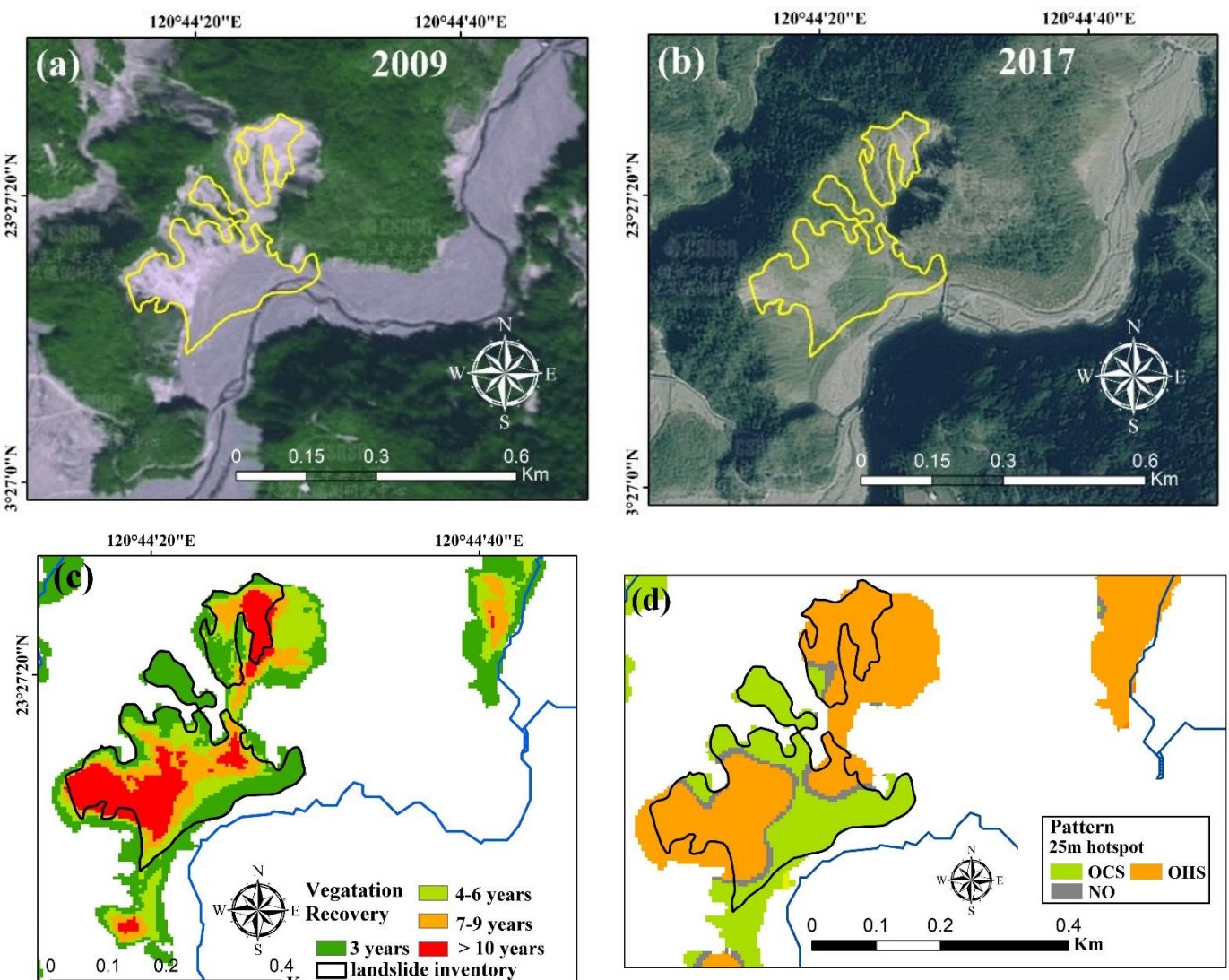

**Figure 8.** The aerial images in 2009 (**a**) and 2017 (**b**), the distribution of vegetation recovery time (**c**), and the spatiotemporal hotspots (**d**) in the riverbank landslide in the T03 subwatershed (Figure 6).

*4.5. Landslide Spatiotemporal Hotspot Distribution and Trend*

The total number of 5 m × 5 m grids in the landslide STC model, collected from the aggregated spatiotemporal data corresponding to all landslide grids from 2001 to 2017 in the *TRW*, was 17,861,645. The spatiotemporal hot spot distribution in the *TRW* is displayed in Figure 9. Neighborhood distance settings in analyses of spatiotemporal landslide hotspots have not yet been suggested. Therefore, these settings were based on a performance comparison of results obtained through analyses performed using various neighborhood distances. Specifically, the distances were set to 5, 25, 50, 100, and 200 m.

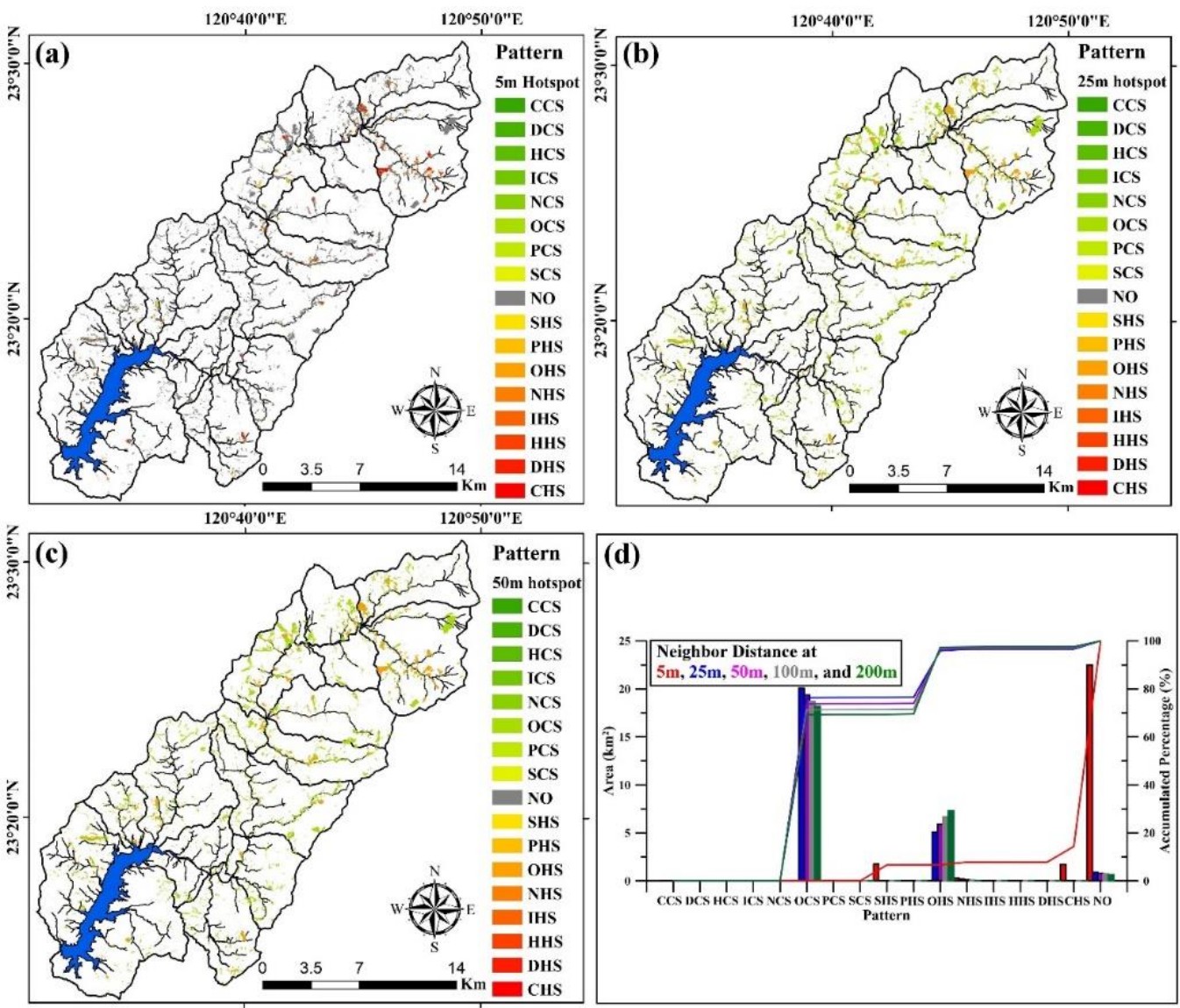

**Figure 9.** The landslide spatiotemporal hotspot distributions (**a**–**c**) and the occupied percentage (**d**) in the *TRW*.

The no pattern area in the analysis with a 5 m neighborhood distance occupied 85.6% of the *TRW*, but that with a neighborhood distance of ≥25 m dropped to 2.4–3.4% (Figure 9d). This result indicates that the 5 m neighborhood distance setting in the analysis was excessively short for detecting landslide hotspots. Furthermore, the 5 m setting yielded only five spatiotemporal patterns for the entire *TRW*. By contrast, when neighborhood distances of ≥25 m were used, 10–12 spatiotemporal patterns were generated (Figure 9a–c). The main hotspot patterns in the results obtained using a 5 m neighborhood distance were sporadic hotspot (SHS) and consecutive hotspot (CHS). The corresponding patterns in the results obtained using a neighborhood distance of ≥25 m were oscillating hotspot (OHS) and oscillating coldspot (OCS). The spatiotemporal distribution of hotspots in the results obtained using 25, 50, 100, and 200 m neighborhood distances were similar yet distinct from that in the results obtained using the 5 m neighborhood distance. Therefore, the neighborhood distance setting in spatiotemporal analyses of landslide hotspots was suggested to be ≥25 m from the result in the study.

The numbers of 5 m × 5 m grids corresponding to coldspot trends in the analyses performed using neighborhood distances of 200 and 25 m were 2.46 and 3.80 times the number of bins corresponding to hotspot trends. These results indicate that the landslide clustering intensity in the *TRW* was low. The summed proportion of OCS and OHS in

analyses with neighborhood distances of ≥25 m in the *TRW* was greater than 95.8%. In other words, the dominant spatiotemporal landslide pattern in the *TRW* was oscillating. The spatiotemporal hot spot pattern of the large landslide in the T01 subwatershed and the riverbank landslide in the T03 subwatershed are presented in Figures 7d and 8d, respectively. Oscillating hotspots and coldspots characterized these two landslides.

## 5. Discussion

The landslide recovery rate and landslide activity are the keys to understanding the evolution of watersheds with frequent and dense landslides. Studies have addressed the rate of landslide recovery induced by large earthquake events [6–8], but few have examined the rate of recovery from landslides induced by extreme rainfall events [19]. The present study discussed and compared differences in the distribution rate of recovery from active landslides induced by large earthquakes and extreme rainfall events.

The majority of landslide types after the 2005 Kashimir earthquake were rock falls and rock slides [6], and those after the 2008 Wenchuan earthquake were debris slides [7]. The majority of landslide types after 2009 Typhoon Morakot in Taiwan were debris slides. Herein, the landslide area decline ratio was defined as the landslide area in the specific year to the landslide area in the year a large earthquake or extreme rainfall event occurred. For example, the landslide areas in the *TRW* in 2009 and 2010 were 1392.9 and 775.6 ha, and the landslide area decline ratio in 2010 was 55.7%. The landslide area decline ratios for the large earthquake- and extreme rainfall-induced landslides are shown in Figure 10. For each landslide event, the linear fitting equation was estimated. The slope coefficient of this equation can be regarded as the average annual landslide area decline ratio, which was used in the comparison of landslide recovery.

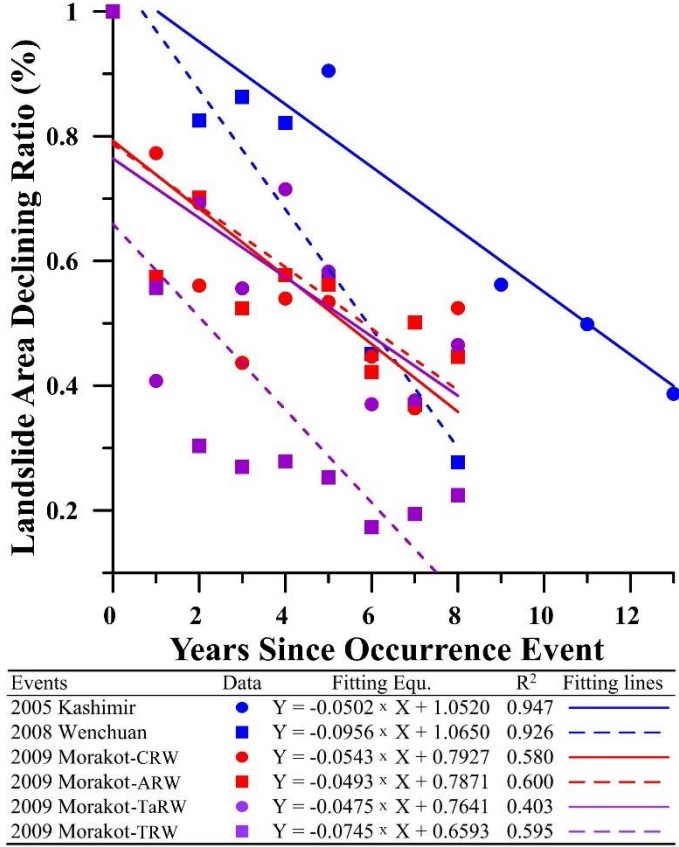

| Events | Data | Fitting Equ. | $R^2$ | Fitting lines |
|---|---|---|---|---|
| 2005 Kashimir | ● | $Y = -0.0502 \times X + 1.0520$ | 0.947 | ——— |
| 2008 Wenchuan | ■ | $Y = -0.0956 \times X + 1.0650$ | 0.926 | – – – |
| 2009 Morakot-CRW | ● | $Y = -0.0543 \times X + 0.7927$ | 0.580 | ——— |
| 2009 Morakot-ARW | ■ | $Y = -0.0493 \times X + 0.7871$ | 0.600 | – – – |
| 2009 Morakot-TaRW | ● | $Y = -0.0475 \times X + 0.7641$ | 0.403 | ——— |
| 2009 Morakot-TRW | ■ | $Y = -0.0745 \times X + 0.6593$ | 0.595 | – – – |

**Figure 10.** The area declining ratio fitting equation of large earthquake-induced and extreme rainfall-induced landslide events.

The annual landslide area decline ratios corresponding to the large earthquake events (5.02–9.56%) were slightly larger than those corresponding to the extreme rainfall events (4.75–7.45%). The $R^2$ values of the linear fitting equations for large earthquake-induced landslides exceeded those of the corresponding equations for extreme rainfall-induced landslides. This indicates that recovery from large earthquake-induced landslides was more stable than recovery was from extreme rainfall-induced landslides. If the declining landslide area in the *TRW* after Typhoon Morakot follows the average annual ratio of landslide area decline, 28.48 years would be required for a return to the pretyphoon landslide area.

The characteristic and evolution comparison of landslide induced by large earthquake events in Northern Pakistan and Sichuan, China and 2009 Typhoon Morakot in *TRW* are shown in Table 5. The main reasons for the difference of landslide evolution between the large earthquake-induced or extreme rainfall-induced landslide cases were the landslide-inducing factors and the deposition location of loose materials yield from landslide. The inducing factor for the earthquake-induced slope instability was the ground shaking force to reduce the shear strength of slope [37], while that for the rainfall-induced slope instability was the infiltration water [38]. The extreme active, very active, and active landslide after large earthquake-induced landslide events were located in the neighboring area of faults, including the Muzaffarbad fault in North Pakistan [6] and the Longmenshan fault in China [7,16], but those after extreme rainfall-induced landslide events in the *TRW* were located in the neighboring area of rivers and gullies (Figure 4). The percentage of the extreme active, very active, and active landslide areas located in the area within 500 m of rivers or the source area of large landslide cases after 2001 Typhoon Nari in the *TRW* was 74.9%, and those after 2009 Typhoon Morakot was 85.3%. The pattern and distribution of large earthquake-induced [6,7] or extreme rainfall-induced landslide spatiotemporal hotspot demonstrated that the landslide hotspot had been moving from mountain regions to the neighborhood area of rivers, but the time needed was the main difference between the large earthquake-induced or extreme rainfall-induced landslide events. The landslide ratio in the area within 500 m to rivers in the first year after 2009 Typhoon Morakot in the *TRW* ranged from 2.4% to 6.3% (Figure 5c), which was much larger than that in the first year after the 2008 Wenchuan earthquake-induced landslide events in China [16]. The evolution of disaster types after extreme rainfall-induced landslide events in the *TRW* was from numerous landslide cases in the hillslope to the river geomorphological changes and riverbank landslide due to much sediment deposition in the rivers, especially in the upstream watersheds. This was also the main reasons for the difficulty of landslide recovery in the T01 and T02 subwatersheds. The key to reduce the disaster in the following years after extreme rainfall-induced landslide events based on the results in the study was how to control the huge amount of loose material yield from numerous landslides and that deposited in the rivers.

**Table 5.** The characteristic and evolution comparison of large earthquake-induced and extreme rainfall-induced landslide events.

| Events | 2005 Kashimir Earthquake | 2008 Wenchuan Earthquake | 2009 Typhoon Morakot |
|---|---|---|---|
| Research area | Northern Pakistan | Sichuan, China | *TRW* in the study |
| Scale | $M_L = 7.6$ | $M_L = 7.9$ | Heavy rainfall with return period greater than 200 years |
| Landslide triggers | ground shaking force to reduce the shear strength of slope | | infiltration water to reduce the shear strength of slope |
| Landslide types | debris fall, debris flow, debris slide, rock falls (majority), rock topple and rock slides (majority) [6] | Rock falls, rock slides, rock flows, debris falls, debris slides (majority), and debris flows [7,16,26] | debris falls, debris slide (majority), and debris flows |
| Sediment deposition location | concentrated in the down-hillslope to rivers near the fault lines | | concentrated in the down-hillslope to rivers |
| Landslide volume (m$^3$) | $2.16 \times 10^9$ [6] | $5–15 \times 10^9$ [39] | $6.65 \times 10^6$ [40] * |

**Table 5.** *Cont.*

| Events | 2005 Kashimir Earthquake | 2008 Wenchuan Earthquake | 2009 Typhoon Morakot |
|---|---|---|---|
| | Distribution of extremely active and very active landslide | | |
| Elevation | 1000m to 1600 m | lower than 2000 m | 1200 m to 1800 m |
| Slope | 30° to 40° | 30° to 50° | 30° to 40° |
| Fault or rivers | close to the Muzaffarbad faults | close to the Longmenshan faults | - |
| Rivers | - | 100 to 400 m to the rivers | <300 m to the rivers |
| | Long-term landslide evolution | | |
| Annual landslide area decline ratios | 5.02% | 9.56% | 7.45% |

*: The landslide volume induced by 2009 Typhoon Morakot in the *TRW* was estimated by using the empirical equations [40].

## 6. Conclusions

This study evaluated landslide activity and vegetation recovery time and detected the spatiotemporal hotspots of extreme rainfall-induced landslides in the *TRW* by using annual landslide inventories and long-term rainfall records from 2001 to 2017. Areas of extreme rainfall-induced landslides and landslides in general decreased consistently following 2001 Typhoon Nari and 2009 Typhoon Morakot. The return period of rainfall events caused a slight increase in the area of number of landslides; during 2002–2008, it was greater than 10 years. In the 2010–2017 period, it was greater than 2 years. The area of extremely active landslides in the *TRW* after 2009 Typhoon Morakot was notably larger than that after 2001 Typhoon Nari, and extremely active landslides were clustered in the subwatersheds in the upper reaches of the *TRW*. The study also discussed the relationship between temporal landslide distribution and geomorphological factors, including elevation, slope, and distance to the river. Landslides in the years following typhoon events were concentrated at 1400 to 2000 m elevations on 35° to 45° slopes within 500 m of the river. The average vegetation recovery time in the *TRW* was 2.29 years, and landslides with vegetation recovery times of greater than 10 years were commonly retrogressive landslide, riverbank landslides in the sinuous reaches, and landslides in the core area of large landslides. The time required to recover from landslides in the subwatersheds in the upper reaches was 1.41 to 1.59 times longer than the time required to recover from landslides in subwatersheds in the middle and lower reaches. The main spatiotemporal pattern of landslides in the *TRW*, including in the subwatersheds in the upper reaches, was characterized by oscillating hotspots and coldspots. The annual landslide area decline ratio in the *TRW* after Typhoon Morakot was estimated to be 94.08%, and approximately 28.5 years is required for a return to the pretyphoon landslide area. Findings on the characteristics of landslide recovery, the distribution of landslide activity, and the spatiotemporal patterns of landslides are useful for watershed management and disaster prevention in areas where rainfall-induced landslides commonly occur.

**Funding:** This research was funded by the Ministry of Science and Technology of Taiwan (R.O.C.), grant number MOST 109-2625-M-035-005 and the Project Research in Feng Chia University (grant number: 20H00710 and 21H00710).

**Institutional Review Board Statement:** Not applicable.

**Informed Consent Statement:** Not applicable.

**Data Availability Statement:** Not applicable.

**Conflicts of Interest:** The authors declare no conflict of interest.

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
