# Peer review of "Evaluating the Landslide Stability and Vegetation Recovery: Case Studies in the Tsengwen Reservoir Watershed in Taiwan"

_water, doi:10.3390/w13243479_

Round 1

Reviewer 1 Report

Dear Author,

I read your paper about landslide evolution mapping in the Tsengwen Reservoir Watershed.

The topic is very interesting, even if it does not fits very well with the scopes of the journal.

I believe that your work is interesting, but there are several issues that must be addressed before it can be considered for publication.

At first, a more sound scientific terminology must be used: what do you mean with headwater landslide, bank-erosion landslide, landslide recovery, etc?

when classifing the type of landslide (and the state of activity), a proper classification must be used; active landslide has a clear meaning, as well as dormant or inactive.

I have some doubt about some results, since you get to some results about rainfall and earthquake induced landslide without making any analysis or using literature work referred to other Countries.

According to what you wrote it looks like that you mix up landslide and erosion process, especially when you wrote about the landslide mapped close to the rivers.

I believe there is a lot to do, before it is worth for publication.

Please find detailed comments in attachment.

Author Response

Dear Water Journal and the reviewers,

My sincere appreciation for the valuable comments from three reviewers. The response for reviewer's comment and the supplement materials were attached in the PDF files.

Chunhung WU

Reviewer 2 Report

This study evaluated the long-term evolution of landslides induced by 2001 Typhoon Nari and 2009  10 Typhoon Morakot in the Tsengwen Reservoir Watershed by using multiannual landslide inventories and rainfall records for the 2001–2017 period.

General comment

The inventory souce data suffers of lack in landslide typology and topological analysis.

See specific comments below:

Title: Authors should explain since title that the "Evaluation" is referred to any type of landslide (shallow, deep-seated, fall, flow, spreading, and so on).
In the text, this consideration is not cited.
For me is very important.

Line 12: Any type of landslide?

Line 19: "loose material" is related to shallow landslides. What type?

Line 78: Please, could you use more appropriate word instead "strata" in designation, i.e. bedrock, or other?

Line 78-81: Provide lithological data about formation without lithostratigraphical attributes.

Line 78-81: Provide lithological data about formation without lithostratigraphical attributes.

Line 109 of the Fig. 1 caption: w of watershed in capital, as previously.

Figure 2 and caption: See comment at Lines 78-81.
The map needs lithostratigraphical and tectonic cross-section.

Line 113: In the upper plot, use a more appropriate representation, discrete, not continous.

Table 1: The criteria about "activity" could be better explained in order to avoid confusion with reactivation of the same landslide.
Different type of landslide have different meaning of activity, e.i. a permanent,  slow-moving, landslide, is Exstremely active?

Table 2: What means "landslide unit"

Line 215: Add a figure with more detailed example of Landslide Activity.

Figure 7c: Legend Heading; Recovery, instead Rocovery 

Line 339: The large landslide cited, what type ? In the aerial images a) and b) different types of landslides are superposed. Please, could you discuss about spatial and topological arrangement, before perform spatial analysis?

Figure 8 c) Recovery instead Ricovery.

Line 364: Different scenarios between 2009 and 2017 are ferred at interaction of fluvial and slope gravitational processes. Please, a geomorphological analysis is required before to perform any statistics. Bank-erosion landslide has to be mapped and distinguished from others landslide types.

Discussion: In the discussion, authors have to discuss about different landslide types between earthquake-induced and rainfall-induced landslide before analysis.

Conclusion have to take into account the previous comments.

References: Different typing characters are used. Please, could authors follow template iournal?

Bet regards

Author Response

(The authors gave the same response as above.)

Reviewer 3 Report

The manuscript does not present a novel study and needs many corrections before considering for publication. please find the comments in the attached PDF.

Author Response

(The authors gave the same response as above.)

Round 2

Reviewer 1 Report

Dear Authors,

thank you for the answer to my comments.

I have few suggestion for you.

You changed the name of headwater landslide into "retrogressive erosion", but according to your description they can be simply called retrogressive landslides. Erosion takes places on the surface and move the material falling from the crown of the landslide, because of the landslide (you should better understand the difference between erosion and landslide).

similarly, bank-erosion landslide, can be called "bank landslide". These phenomena are well known and one of the main causes is the erosion of the landslide toe by the river, so there is not the need of creating new nouns. In this case the river erosion is the triggering factor of the landslide; there are hundreds of examples around the world.

I suggest you a throughout revision of the text, since several sentences need improvement, e.g.:

"Most of the landslide cases in the TRW were shallow landslide cases based on the field investigation result"

can be rewritten in a more clear manner:

"According to the field survey results, the majority of landslides can be classified as shallow landslide."

Best Regards

Author Response

Appreciation for the comments and our response for the comment is attached in the file.

Reviewer 2 Report

I have appreciated the Author's efforts to improve the statistical analysis by a critical pre-processing on landslide data.

Quality is better than quantity in landslide studies.

Just as a kindly suggestion on this topic in the revised text.

Author Response

(The authors gave the same response as above.)

Reviewer 3 Report

I appreciate the authors for their effort in revising the manuscript. Please add the revised text or figures also along with the responses to comments.

Find the comments for the revised version in he attached PDF.

Author Response

(The authors gave the same response as above.)
